# Impact of Sirolimus versus Mycophenolate Mofetil on Kidney Function after Calcineurin Inhibitor Dose Reduction in Liver Transplant Recipients

**DOI:** 10.3390/ph16081087

**Published:** 2023-07-31

**Authors:** Heng-Yi Chiang, Lung-Chih Li, Chien-Ning Hsu, Chih-Che Lin, Yi-Chia Chan, Chih-Chi Wang, Chao-Long Chen

**Affiliations:** 1Division of Nephrology, Department of Internal Medicine, Kaohsiung Chang Gung Memorial Hospital and Chang Gung University College of Medicine, Kaohsiung 833, Taiwan; a055925@cgmh.org.tw; 2Institute for Translational Research in Biomedicine, Kaohsiung Chang Gung Memorial Hospital and Chang Gung University College of Medicine, Kaohsiung 833, Taiwan; 3Department of Pharmacy, Kaohsiung Chang Gung Memorial Hospital and Chang Gung University College of Medicine, Kaohsiung 833, Taiwan; chien_ning_hsu@hotmail.com; 4School of Pharmacy, Kaohsiung Medical University, Kaohsiung 833, Taiwan; 5Liver Transplant Center, Department of Surgery, Kaohsiung Chang Gung Memorial Hospital and Chang Gung University College of Medicine, Kaohsiung 833, Taiwan; immunologylin@gmail.com (C.-C.L.); sparkle@cgmh.org.tw (Y.-C.C.); ufel4996@gmail.com (C.-C.W.); clchen@cgmh.org.tw (C.-L.C.)

**Keywords:** sirolimus, mycophenolate mofetil, liver transplantation, acute kidney disease, chronic kidney disease

## Abstract

Impaired kidney function is associated with increased morbidity and mortality in patients undergoing liver transplantation. Although immunosuppressants are essential in these patients, they impair kidney function. This study aimed to compare adverse kidney outcomes between patients treated with a reduced dose of tacrolimus (calcineurin inhibitor) plus sirolimus or mycophenolate mofetil (MMF) in the liver transplant center at Kaohsiung Chang Gung Memorial Hospital between April 2011 and December 2017. Propensity score matching was used to identify 232 patients. The risk of adverse kidney outcomes was estimated using Cox proportional hazards regression, and changes in kidney function over time were analyzed using linear mixed modeling. Acute kidney disease risks in this study cohort were not significantly different for the two immunosuppressants (aHR 1.04; 95% CI: 0.70–1.55, *p* = 0.8328). However, sirolimus use was significantly associated with a higher risk of estimated glomerular filtration rate decline > 30% than MMF (aHR, 2.09; 95% CI: 1.33–3.28; *p* = 0.0014). Our results demonstrate that sirolimus use may have worsened long-term kidney outcomes compared to MMF. Close monitoring of kidney function, dose adjustment, and timely transition to MMF is necessary for LT patients receiving sirolimus.

## 1. Introduction

Liver transplantation (LT) is an effective treatment for end-stage liver disease [1]. The discovery of calcineurin inhibitors (CNIs), including cyclosporine and tacrolimus, has revolutionized the long-term prognosis of organ transplants by preventing graft rejection [2]. However, these agents can have significant side effects, of which acute or chronic nephrotoxicity are considered major contributors to chronic kidney disease (CKD) in liver transplant recipients [3]. In previous studies, up to 20% of the recipients developed stage 4 or 5 CKD within 20 years of LT [4,5].

Kidney function plays an important role in the long-term outcomes of patients undergoing LT. Both acute kidney injury (AKI) and CKD, which are associated with short- and long-term risks, develop after LT. AKI is a condition defined by a sudden loss of kidney function on the basis of increased serum creatinine levels or reduced urinary output within a duration of 7 days [6]. CKD is defined as an abnormality of kidney structure or function for more than 3 months [7]. The development of post-LT AKI is associated with an increased risk of CKD [8], prolonged hospital stay [9], higher incidence of graft failure [10], and even higher mortality rates [8,11,12]. CKD is the final manifestation of persistent intrinsic renal damage, particularly after several AKI events. Further, CKD after LT is associated with an increased risk of death [3,4,13], graft failure [3], and other adverse outcomes, such as cardiovascular diseases [14,15]. Acute kidney disease (AKD), defined as kidney damage lasting between 7 and 90 days after AKI, is regarded as a continuum between AKI and CKD [16,17]. Despite many studies on LT reporting AKI and CKD, very few have investigated AKD.

Post-LT immunosuppressive agents, particularly CNIs, play important roles in the development of post-LT AKI and CKD. Possible mechanisms of CNI nephrotoxicity involve afferent and efferent glomerular arteriole vasoconstriction, increased platelet aggregation, and the subsequent development of thrombotic microangiopathy within the glomerular arterioles [18]. To minimize the nephrotoxicity of CNIs, reduced doses of these agents, in combination with mammalian target of rapamycin inhibitors (mTORi) or mycophenolic acid (MPA), are commonly used in LT recipients [19].

Sirolimus, introduced in the 1990s, was the first mTORi and has been widely used in many liver transplant centers, including ours. It inhibits the mTOR pathway, which is involved in the activation and proliferation of immune cells. Earlier studies have demonstrated that either sirolimus monotherapy or sirolimus plus a reduced dose of a CNI could offer renal protection from CNI-induced nephrotoxicity [20,21]. Hence, although the FDA issued a black box warning for increased hepatic artery thrombosis and graft loss in de novo liver transplant recipients [22], many centers still use sirolimus to spare CNI use but avoid de novo use immediately after LT.

Mycophenolate mofetil (MMF), a semi-synthetic 2-morpholinoethyl ester of MPA, is a selective and reversible noncompetitive inhibitor of inosine monophosphate dehydrogenase, a crucial enzyme in DNA formation [23]. MMF is often used in combination with other immunosuppressive medications to prevent the rejection of transplanted organs and reived FDA approval for use in LT in 2000 [23]. Many studies have shown that MMF is a safe immunosuppressant that offers metabolic and renal benefits over the standard dose of CNIs [24]. In a multicenter, randomized controlled study, Boudjema et al. showed that reduced-dose tacrolimus with MMF decreased the incidence of renal dysfunction compared with standard-dose tacrolimus [25]. Another prospective multicenter randomized study showed that the introduction of MMF combined with a reduced CNI dose of at least 50% resulted in a significant improvement in renal function [26].

Previous studies have demonstrated that CNI dose reduction combined with sirolimus or MMF contributes to the preservation of kidney function and is a useful strategy for reducing the risk of nephrotoxicity in LT recipients. However, kidney function deterioration still occurs in these patients; therefore, a head-to-head comparison between sirolimus and MMF combined with a CNI is warranted. Consequently, this study aimed to examine the effects of sirolimus and MMF on kidney outcomes in patients undergoing LT.

## 2. Results

### 2.1. Pre-Operation Baseline Patient Characteristics

Between April 2011 and December 2017, 789 patients underwent liver transplantation (LT) at Kaohsiung Chang Gung Memorial Hospital (KCGMH). We identified 472 patients who met the inclusion criteria (318 in the sirolimus group and 154 in the MMF group). Notably, 69 patients were excluded due to a follow-up period of less than 3 years, of which 41 were foreign nationals and could not be consistently monitored at our center. Furthermore, 74 patients were excluded because they did not use sirolimus or MMF as their immunosuppressant. An additional 52 patients were excluded as they were treated with counterpart therapy during the follow-up period. Finally, five patients were excluded as they used the target agents for less than a year (Figure 1). Prior to propensity score (PS) matching, new sirolimus users were significantly older than new MMF users (54.27 vs. 51.66 years old, *p* = 0.02). The majority of indications for LT in our cohort are hepatitis B, hepatitis C-virus-related end-stage liver disease, and hepatocellular carcinoma (HCC). The proportion of patients with hepatocellular carcinoma (HCC) (48.4% vs. 28.6%, *p* < 0.001) and baseline proteinuria (12.3% vs. 5.8%, *p* value = 0.03) were higher among sirolimus users than among MMF users. Patients in the sirolimus group had a lower baseline estimated glomerular filtration rate (eGFR) than those in the MMF group (86.9 vs. 100.4 mL/min/1.73 m^2^, *p* < 0.001). The proportion of patients with a baseline eGFR < 60 mL/min/1.73 m^2^ was higher in the sirolimus group than in the MMF group (16.7% vs. 1.9%) (Table 1).

The baseline characteristics of the study cohort are presented in Table 1. The PS-matched groups each consisted of 116 patients with an average age of 52.3 years. The demographic and clinical characteristics of sirolimus and MMF users were well balanced (standardized mean difference (SMD) < 0.1). Few primary biliary cholangitis (PBC) was noted in the two groups (three patients in the sirolimus group and five in the MMF group (*p* = 0.36)). Neither nonalcoholic steatohepatitis (NASH) nor primary sclerosing cholangitis (PSC) were observed in our study cohorts.

### 2.2. Intra- and Post-Operative Factors

Certain intra- and post-operative factors, such as blood loss, cold and warm ischemic time, and post-op acute rejection rate, may influence renal outcomes after liver transplantation [18,27]. There was no significant statistical difference between the two groups in terms of blood loss (sirolimus vs. MMF group: 3363 ± 3702 vs. 3178 ± 3720 mL, *p* = 0.71), warm ischemia time (sirolimus vs. MMF group: 42.21 ± 29.01 vs. 39.57 ± 18.85 min, *p* = 0.41), and cold ischemia time (sirolimus vs. MMF group: 42.47 ± 14.13 vs. 41.72 ± 11.09 min, *p* = 0.65) during surgery.

Liver function one month after LT was also compared between the two groups. There was no significant statistical difference in aspartate aminotransferase (AST) (sirolimus vs. MMF group: 42.49 ± 35.67 vs. 40.16 ± 27.71 U/L, *p* = 0.58), alanine transaminase (ALT) (sirolimus vs. MMF group: 66.87 ± 74.83 vs. 67.46 ± 62.98 U/L, *p* = 0.95), total bilirubin (sirolimus vs. MMF group: 0.96 ± 1.66 vs. 0.83 ± 0.42 mg/dL, *p* = 0.42), and albumin (sirolimus vs. MMF group: 3.74 ± 0.52 vs. 3.75 ± 0.44 g/dL, *p* = 0.41) between the two groups. The number of biopsy-proven acute rejection occurring in two months after LT were 15 in the sirolimus group (12.9%) and 11 (9.4%) in the MMF group (*p* = 0.40).

### 2.3. Patterns of Immunosuppressant Treatment

The initiation of sirolimus treatment in the sirolimus group varied based on postoperative conditions, with a mean value of 10.7 ± 3.1 days after LT. The mean sirolimus dose was 1.60 ± 0.70 mg/day. In the MMF group, the initiation of MMF treatment occurred 2.4 ± 0.5 days after LT, and the mean dose of MMF was 944 ± 167 mg/day. For both the sirolimus and MMF groups, tacrolimus treatment was initiated according to the protocol described in Section 4. To investigate the effect of CNIs on nephrotoxicity, tacrolimus trough levels were recorded at the index date and 6 months after tacrolimus initiation. On the index date, tacrolimus trough levels in the MMF group were higher than in the sirolimus group (dose of tacrolimus in sirolimus vs. MMF groups: 2.55 ± 1.52 vs. 2.96 ± 0.98 mg/day, *p* < 0.001; trough levels: 4.51 ± 1.87 vs. 5.49 ± 1.73 ng/mL, *p* < 0.001).

At 6 months after initiation, the doses of tacrolimus were higher in the MMF group than in the sirolimus group (sirolimus vs. MMF group: 2.13 ± 1.51 vs. 2.78 ± 0.87 mg/day, *p* < 0.001). The trough levels of tacrolimus were also higher in the MMF group than in the sirolimus group (sirolimus vs. MMF group: 3.51 ± 2.18 ng/mL vs. 4.89 ± 2.51 ng/mL, respectively, *p* < 0.001), both of which were within the CNI low dose range [2].

### 2.4. Adverse Kidney Outcomes

Over the course of the 3-year follow-up duration, the observed cumulative occurrences of AKD stood at 47.4% (N = 55) and 50% (N = 58) for the sirolimus and MMF cohorts, respectively (*p* = 0.693). In terms of the cumulative occurrences of advanced AKD, they were recorded as 11.2% (N = 13) and 10.3% (N = 12) in the sirolimus and MMF group, respectively (*p* = 0.832). A reduction to 30% of the baseline value in eGFR was found in 56.0% (N = 65) of the sirolimus cohort and 34.5% (N = 40) of the MMF group (*p* = 0.001). Moreover, a reduction to 50% of the baseline value in eGFR occurred in 19.8% (N = 23) of the sirolimus cohort and 3.5% (N = 4) of the MMF cohort (*p* = 0.0001) (Table 2).

### 2.5. Acute Kidney Disease

As shown in Figure 2a, there was no significant difference in the incidence of AKD between the two treatment groups (log-rank test, *p* = 0.67). In addition, no significant difference was observed in patients with advanced (stage 2, 3) AKD (Figure 2b) (log-rank test, *p* = 0.83). Cox regression analysis also showed that sirolimus was not significantly linked to an increased risk of AKD compared to MMF (aHR, 1.04; 95% CI: 0.70–1.55, *p* = 0.83). In subgroup analysis, there were also no significant differences between the two groups in terms of age, sex, baseline renal function, or presence of comorbidities, including HBV, HCV, HCC, or DM (Figure 3a).

### 2.6. eGFR Decline from Baseline

Decreases in estimated GFR smaller than a doubling of serum creatinine concentration are strongly linked to the risk of End-Stage Renal Disease (ESRD) and mortality, supporting the consideration of lesser declines in estimated GFR (such as a 30% reduction over 2 years) as an alternative endpoint for CKD progression [28]. Moreover, a 50% decline in eGFR is often regarded as a major indicator of adverse kidney events in many significant studies [29,30]. Figure 2c illustrates that over the course of this study, the cumulative incidence of eGFR decline greater than 30% was significantly elevated in the sirolimus group compared to the MMF group (log-rank test, *p* = 0.0004). A more prominent difference was evident between the two groups in the cumulative events of eGFR decline of over 50% (log-rank test, *p* < 0.0001) (Figure 2d). Further, the Cox regression analysis showed a notable association between the use of sirolimus and a higher risk of a 30% reduction in eGFR, compared to the use of MMF (aHR, 2.09; 95% CI: 1.33–3.28, *p* < 0.0014). In the subgroup analysis, we found that patients with the following characteristics had a significantly higher risk of a 30% decline of eGFR in the sirolimus group than in the MMF group: male, age ≥ 55, baseline EGFR ≥ 90, hepatitis B virus (HBV), no hepatitis C virus (HCV), or HCC, non-DM, and AKD events during the follow-up period (Figure 3b).

### 2.7. Changes in eGFR over Time

As shown in Table 3, the median follow-up (6 months) decline in eGFR was greater in the sirolimus group than in the MMF group. This difference was statistically significant (adjusted β = −0.18; 95% CI, −0.3–0.06 mL/min/1.73 m^2^, *p* = 0.004) after adjusting for other factors, such as age, baseline eGFR, model for end-stage liver disease (MELD) score, tacrolimus trough level at both the index date and 6 months, and baseline comorbidities (proteinuria, HBV, HCV, and HCC).

## 3. Discussion

In this study, we found that patients receiving sirolimus had a similar risk of developing AKD as those receiving MMF. However, sirolimus had a higher risk of >30% decline in eGFR than MMF. MMF offered a protective effect against a 30% decline in eGFR compared to sirolimus, particularly for male patients, those aged over 55 years, those with a better baseline eGFR (>90 mL/min/1.73 m^2^) and AKD, and patients without HCC, HCV, or DM. To the best of our knowledge, this is the first study to compare kidney outcomes after sirolimus and MMF administration in patients undergoing LT.

Our study found that sirolimus and MMF did not significantly differ in their impact on the risk of developing AKD after LT. AKI or AKD is a crucial prognostic risk factor following LT, and its occurrence may be linked to increased mortality [8,11,12], extended hospital stay, higher hospitalization expenses [9], graft failure [10], and persistent renal dysfunction post-LT [8]. Moreover, the current discussions on AKD are relatively limited. In previous studies, the incidence of post-LT AKD was reported in more than 50% of patients, with 15% requiring renal replacement therapy, which is similar to the findings of our study (sirolimus: 47.41% and MMF: 50%) [14,31]. Pre-transplant renal dysfunction is an important risk factor for post-LT AKI [31,32]. As the baseline renal function in our cohort was relatively preserved (98.3% with eGFR > 60 mL/min/1.73 m^2^), both sirolimus and MMF demonstrated similar effects on short-term renal function. There were also no significant differences between the two groups in terms of age, sex, baseline renal function, or presence of comorbidities, including HBV, HCV, HCC, or DM, in the subgroup analysis (Figure 3a).

CKD, defined as a reduced eGFR for ≥3 months, is a common complication that has a major impact on graft and patient survival in LT [4,33]. A decline in eGFR of 30% has been proven to correlate well with the development of CKD and can be used to predict death and advanced CKD requiring dialysis in patients who undergo LT [28,34,35]. Many studies have indicated CNIs as the main cause of CKD in LT recipients [3]. MMF and mTORi are used as strategies for renal protection to minimize the requirement for high CNI doses [2]. However, head-to-head comparisons between MMF and sirolimus are scarce. Our study showed that MMF had a lower incidence of eGFR decline of >30% than sirolimus (aHR, 2.09; 95% CI: 1.33–3.28, *p* < 0.0014) after 3 years of follow up (Figure 3b). Although tacrolimus exposure is an important factor in nephrotoxicity after LT, tacrolimus trough levels change dynamically and are difficult to record. The tacrolimus trough at the index date and 6 months after were used as confounders for EGFR decline > 30%. The reason why the initial dose and trough of tacrolimus in the sirolimus group at the index date were lower than those in the MMF group may be because sirolimus was given later than MMF (10.7 ± 3.1 vs. 2.4 ± 0.5 days). Six months after initiation, the tacrolimus dose is considered stable [36,37]. It has been reported that sirolimus reduces the tacrolimus dose owing to pharmacokinetic alterations [38,39], which may explain why both the doses and trough levels of tacrolimus at 6 months were higher in the MMF group than in the sirolimus group. However, the higher tacrolimus trough level in the MMF group did not cause worse renal outcomes, suggesting that tacrolimus was not the reason for the inferior renal outcomes in the sirolimus group. Previous studies have reported that renal function in the first and second years after LT may predict renal function 5 years later [40,41]. Therefore, the results from this current study may have potential value as references for long-term renal function prediction in LT patients.

Although no significant difference in AKD incidence was observed between the sirolimus and MMF groups, LT patients with a history of AKD were more likely to have worse long-term renal function compared with those treated with MMF (Figure 3b). The mechanisms by which sirolimus is associated with poor renal outcomes remain unclear. DuBay et al. illustrated that LT patients with an initial creatinine clearance < 30 mL/min who were switched to sirolimus were worse off than patients maintained on low-dose CNI [42]. Li et al. reported that LT patients with proteinuria and poor baseline renal function had poor renal outcomes and survival [43]. It has also been reported that sirolimus may be associated with worse allograft survival in patients undergoing kidney transplants [44]. In this study, patients who received sirolimus exhibited a higher >30% decrease in eGFR. Although no significant differences were observed in some subgroup analyses, this could be due to the small numbers in certain subgroups. Nevertheless, most subgroup analyses showed a protective effect against MMF. Notably, the protective effects of MMF were most pronounced in older male patients who experienced AKD during the follow-up period (Figure 3b).

Some limitations of our study should be acknowledged. First, this was a retrospective study conducted at a single medical center. Second, we selected patients who underwent follow up for at least 3 years (of the 69 patients who were lost to follow up, 41 were foreign nationals), which ensured that the data were complete at different time points in the follow-up period but may not be representative of all liver transplant patients at our medical center. Third, we acknowledge that unmeasured confounders, such as volume status, smoking, socioeconomic status, and donor factors, associated with AKD and CKD development, were not assessed. However, we attempted to minimize these differences via PS matching to ensure that the two patient groups had similar basic characteristics. Our study also has strengths. Being conducted at one of the largest LT centers, the number of cases in this study was sufficiently large for successful PS matching. Additionally, the follow-up period was representative of long-term renal function. To the best of our knowledge, this is the first study to compare the short- and long-term renal outcomes between sirolimus and MMF plus reduced-dose tacrolimus in LT patients. These study results highlight the need for tailored immunosuppression for LT patients who are at risk of kidney disease. The use of sirolimus is not preferable for patients with existing kidney disease such as previous AKD and MMF combination may be a better strategy for long-term kidney protection. Monitoring kidney function and trough levels of tacrolimus and sirolimus is imperative for the protection of kidney function. Furthermore, these study findings can serve as platforms for future research comparing the effectiveness and safety of different strategies of immunosuppression.

## 4. Materials and Methods

### 4.1. Patients and Study Design

This retrospective cohort study enrolled living-donor liver transplant recipients between April 2011 and December 2017 at Kaohsiung Chang Gung Memorial Hospital, Kaohsiung, Taiwan. We routinely injected basiliximab (20 mg), a CD25 monoclinal antibody, 6 h after portal vein reperfusion and on post-transplant day 4 for induction therapy. The implementation of basiliximab has been improved with reduced biopsy-proven acute rejection rates during the initial 6 months and improved problem-free post-LT survival over the first 12 months. Additional advantages of basiliximab include the preservation of kidney function, reduction in steroid-resistant acute rejection occurrences, and lower incidence of post-transplant diabetes mellitus [45]. Administration of tacrolimus is postponed until improvement in renal function. Upon achieving these markers, oral tacrolimus administration is initiated at 0.15 mg/kg/day. Dosage adjustments are made to achieve a trough level of 10–15 ng/mL during the first week, >6 ng/mL beyond the first week, and >4 ng/mL after the first year, or even <4 ng/mL, provided there is normal liver function [46]. Both sirolimus and MMF were administered as tacrolimus dose-reducing regimens. Sirolimus was used for patients diagnosed with HCC [40] in the first 4.5 years (2011–2015) until Geissler et al. published in 2016 that sirolimus in LT recipients with HCC did not improve long-term recurrence-free survival beyond 5 years [47]. Sirolimus was initiated at 1 mg once daily for 3–7 days and then titrated to main a trough level of 4–10 ng/m [48]. The dose of MMF used was initiated at 1–2 gm/day. Methylprednisolone (20 mg/kg IV) was administered, followed by a post-transplantation IV dosage of 2 mg/kg/day. This was gradually reduced to an oral prednisone dosage of 20 mg daily by the seventh postoperative day, eventually achieving a minimum dosage of 5 mg per day. From the third month post-transplantation, we began weaning patients off steroids unless they have experienced rejection episodes or if the transplantation was performed due to an autoimmune disease.

In order to exclusively consider new users of sirolimus and MMF, we excluded patients who did not have medical records extending at least a year prior to the initiation of treatment. Additionally, those who had been administered counteracting therapy at any point during the follow-up period were also excluded. The date of sirolimus or MMF treatment initiation was defined as the study index date. Patients under 18 years of age or with a follow-up period of less than 3 years (e.g., loss to follow-up or mortality) were excluded. Other exclusion criteria were the non-use of sirolimus or MMF as immunosuppressants or the use of target immunosuppressants for less than 1 year.

Post-LDLT patients are monitored in an outpatient setting every 1–3 months. Dosages of immunosuppressants are adjusted accordingly. Additional ancillary tests such as CT scans and MRI/MRCP are also conducted based on the initial signs, symptoms, and results of other routine tests [46].

The study protocol was approved by the Institutional Review Board and Ethics Committee of the Chang Gung Medical Foundation, Taoyuan, Taiwan (IRB No. 202202009BO), and adhered to the principles of the Declaration of Helsinki and Declaration of Istanbul.

### 4.2. Outcomes

The outcomes of interest were AKD, advanced AKD (stage 2 or stage 3 AKD), and 30% and 50% estimated glomerular filtration rate (eGFR) decline from baseline [28]. AKD was defined as an increase in serum creatinine level to ≥1.5 times, and advanced AKD (stage 2 and stage 3 AKD) was defined as an increase in serum creatinine level to ≥2.0 times that occurred or persisted within 90 days after renal injury [16]. The reason for using AKD instead of AKI was that most LT patients were followed up at the outpatient clinic with renal function monitoring intervals ranging from 7 to 90 days. Furthermore, there is currently limited research on the potential occurrence of AKD in LT patients due to these two medications and the impact of AKD occurrence on the prognosis of LT patients.

The eGFR was calculated using the Taiwan version of the abbreviated Modification of Diet in Renal Disease (MDRD) equation (eGFR = 175 × (Scr/88.4)^−1.154^ × (Age)^−0.203^ × (0.742 if female)) [49], which is the daily practice method for monitoring kidney function in Taiwan [49]. The time interval averages of eGFR at each time interval (6 months) were used to assess the change in eGFR over time (baseline eGFR and last averaged eGFR) between the treatment groups.

### 4.3. Covariates

The following study covariates were measured at baseline as potential confounders to be adjusted in the analyses: age, sex, body mass index (BMI), MELD score, baseline comorbidities [hypertension, HBV, HCV, HCC, diabetes mellitus (DM), alcoholism and proteinuria (defined as dipstick urinalysis protein ≥ 1+)], baseline biochemical data (eGFR, albumin, Hb), and other medications with ≥28 days of use [ACEI/ARB, non-steroid anti-inflammatory drugs (NSAID), statins, and diuretics]. Trough levels of tacrolimus at the index date and 6 months after (considered a relatively stable dosage period) were considered potential confounders for kidney function change from baseline (Appendix A).

### 4.4. Statistical Analysis

To level the baseline demographic and clinical discrepancies between the sirolimus and MMF groups, PS matching was used. The individual propensity scores of patients treated with either sirolimus or MMF were approximated via logistic regression. We matched new users of sirolimus and MMF on a 1:1 basis using the greedy technique in the PS-matching method. Before and after PS matching, we evaluated baseline traits and comorbidities between the two treatment groups. The standardized mean difference (SMD) was used to gauge comparability in baseline traits between groups, with an SMD < 0.1 considered as no meaningful difference.

We utilized the chi-squared test to calculate adverse kidney outcomes (AKD and eGFR decline). The time-to-adverse kidney outcome endpoint was probed using the Kaplan–Meier approach with log-rank tests. We employed the Cox proportional hazard regression model to scrutinize the independent correlation between the selection of MMF (or sirolimus) and the occurrence of individual adverse kidney diseases. A linear mixed-effects model was employed to estimate the discrepancy in the average eGFR changes over time between sirolimus and MMF users. To evaluate the heterogeneous impacts of sirolimus and MMF by varying baseline characteristics, we conducted stratified analyses in the matched cohorts considering sex, age (<55 years vs. >55 years), and baseline eGFR groups (<90, and ≥90 mL/min/1.73 m^2^), HBV, HCV, HCC, DM, and AKD in eGFR decline. A two-tailed test result of *p* < 0.05 was considered statistically significant. All statistical analyses were performed using SAS 9.4 (SAS Institute, Cary, NC, USA).

## 5. Conclusions

In conclusion, our study showed that among LT patients, the use of sirolimus plus tacrolimus may carry a higher risk of long-term kidney function deterioration than the use of MMF plus tacrolimus. These findings highlight the need for periodical monitoring of kidney function and early detection of kidney injury at least every 1–3 months and other reno-protective strategies, such as dose optimization of immunosuppression and early transition to MMF in LT patients using sirolimus.

## Figures and Tables

**Figure 1 pharmaceuticals-16-01087-f001:**
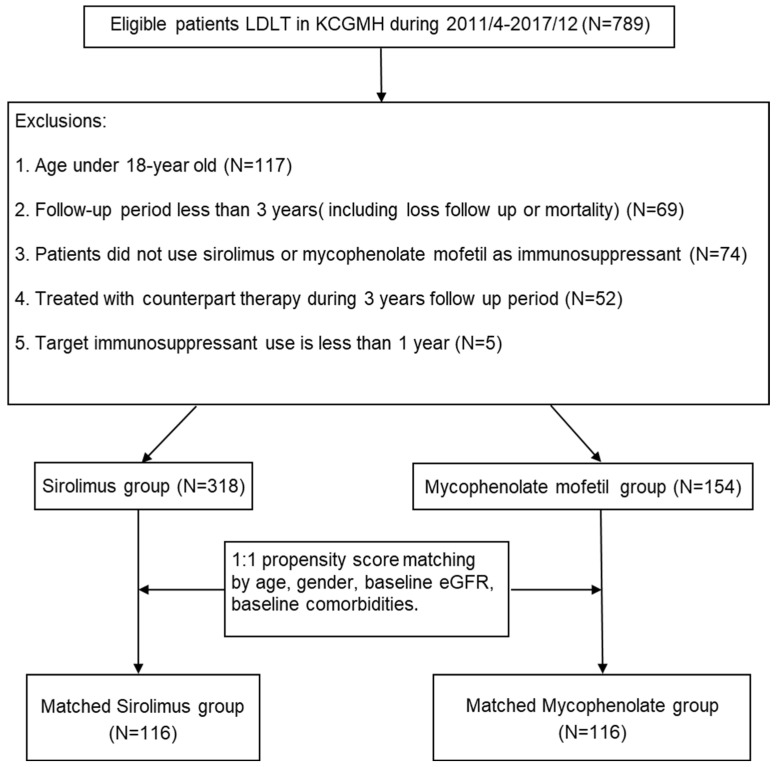
Flowchart and patient selection process. Abbreviations: LDLT, living donor liver transplant; KCGMH.

**Figure 2 pharmaceuticals-16-01087-f002:**
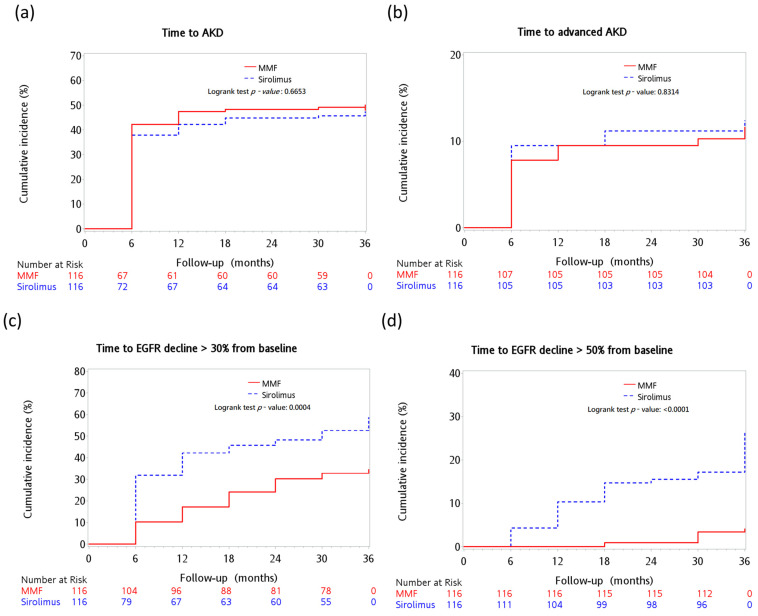
Cumulative incidence of kidney outcomes in the sirolimus and MMF groups. (**a**) AKD event (log-rank test, *p* < 0.76); (**b**) advanced AKD event (log-rank test, *p* = 0.84); (**c**) eGFR decline of >30% from baseline (log-rank test, *p* ≤ 0.001); (**d**) eGFR decline of >50% from baseline (log-rank test, *p* ≤ 0.001) (N = 232). Abbreviations: MMF, mycophenolate mofetil.

**Figure 3 pharmaceuticals-16-01087-f003:**
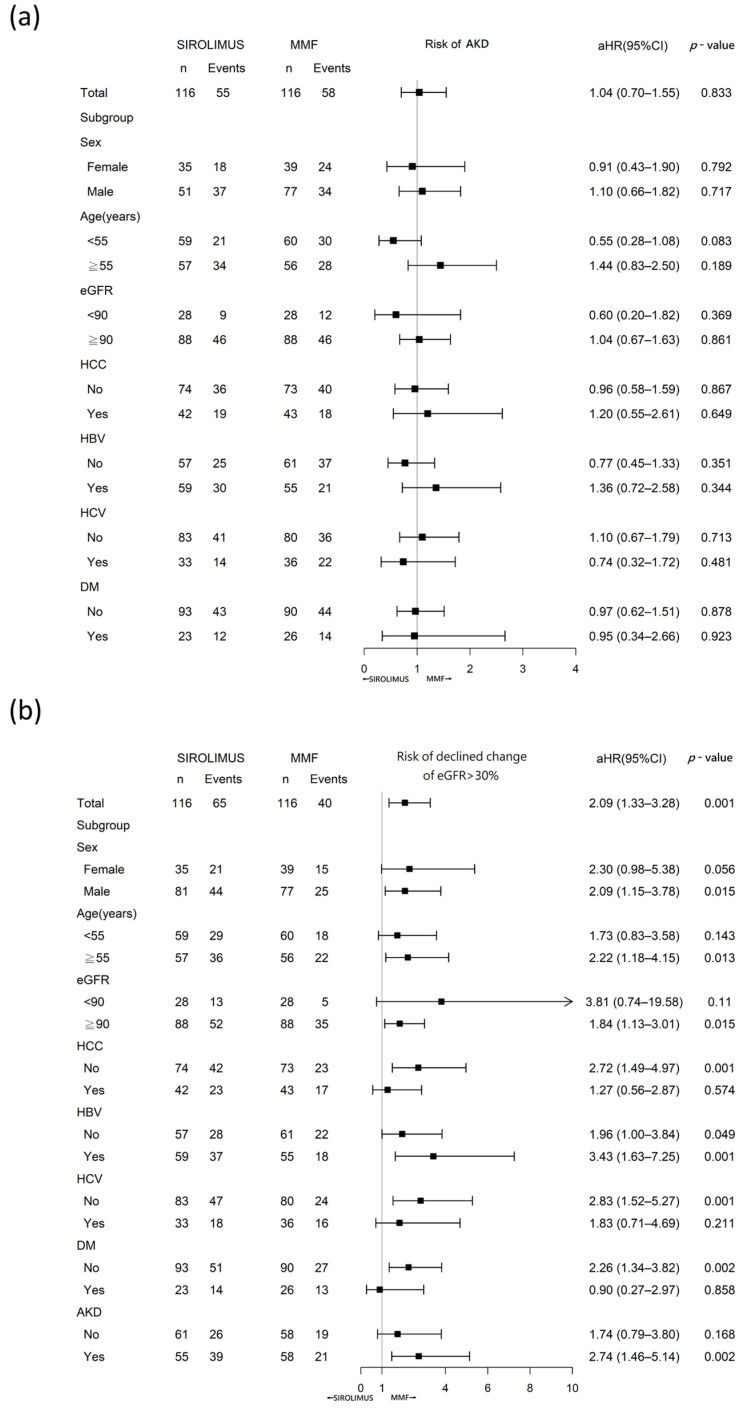
Cox regression model of (**a**) AKD and (**b**) eGFR decline of >30% from baseline between sirolimus and MMF groups. Abbreviations: DM, diabetes mellitus; MMF, mycophenolate mofetil; aHR, adjusted hazard ratio.

**Table 1 pharmaceuticals-16-01087-t001:** Characteristics of the study cohort.

	Before PSM			After PSM		
	Sirolimus	MMF	SMD	Sirolimus	MMF	SMD
	N = 318	N = 154		N = 116	N = 116	
Age (mean (SD))	54.3 (8.6)	51.7 (10.5)	0.27	52.3 (10.2)	52.3 (9.5)	0.004
eGFR, mL/min/1.73 m^2^ (mean (SD))	86.9 (25.4)	100.4 (15.2)	0.57	97.1 (18.3)	100.2 (14.9)	<0.001
eGFR < 60	53.0 (16.7)	3.0 (1.9)		2.0 (1.7)	2.0 (1.7)	
eGFR 60–89.9	85.0 (26.7)	33.0 (21.4)		26.0 (22.4)	26.0 (22.4)	
eGFR ≧ 90	180.0 (56.6)	118.0 (76.6)		88.0 (75.9)	88.0 (75.9)	
Hypertension (%)	67.0 (21.1)	24.0 (15.6)	0.14	13.0 (11.2)	16.0 (13.8)	0.078
Diabetes mellitus (%)	81.0 (25.5)	35.0 (22.7)	0.06	23.0 (19.8)	26.0 (22.4)	0.063
Male sex (%)	248.0 (78.0)	104.0 (67.5)	0.24	81.0 (69.8)	77.0 (66.4)	0.074
HBV (%)	140.0 (44.0)	69.0 (44.8)	0.02	59.0 (50.9)	55.0 (47.4)	0.069
HCV (%)	78.0 (24.5)	59.0 (38.3)	0.30	33.0 (28.4)	36.0 (31.0)	0.057
HCC (%)	154.0 (48.4)	44.0 (28.6)	0.42	42.0 (36.2)	43.0 (37.1)	0.018
baseline proteinuria > 1+ (%)	39.0 (12.3)	9.0 (5.8)	0.23	8.0 (6.90)	9.0 (7.8)	0.033
MELD score (mean (SD))	13.8 (7.6)	13.5 (6.9)	0.05	14.1 (7.6)	13.6 (7.3)	0.071
BMI (mean (SD))	25.55 (3.9)	24.5 (4.3)	0.25	25.0 (4.0)	24.9 (4.5)	0.025
baseline albumin (mean (SD))	3.2 (0.60)	3.06 (0.61)	0.28	3.1 (0.5)	3.2 (0.6)	0.037
baseline Hb (mean (SD))	10.5 (2.3)	10.3 (2.2)	0.10	10.6 (2.2)	10.4 (2.3)	0.099

Abbreviations: PSM, propensity score matching; SMD, standardized mean difference; HBV, hepatitis B virus; HCV, hepatitis C virus; HCC, hepatocellular carcinoma; MELD, model for end-stage liver disease; BMI, body mass index; Hb, hemoglobin.

**Table 2 pharmaceuticals-16-01087-t002:** Adverse kidney outcomes.

Outcomes	Event N	Sirolimus (N = 116)	MMF (N = 116)	*p* Value
		N(%)	N(%)	
AKD	114	55 (47.4)	58 (50)	0.693
advanced AKD	25	13 (11.2)	12 (10.3)	0.832
EGFR decline > 30% from baseline	105	65 (56.0)	40 (34.5)	0.001
EGFR decline > 50% from baseline	27	23 (19.8)	4 (3.5)	0.0001

AKD is defined as an increase in serum creatinine level to ≥1.5 times and advanced AKD (stage 2 and stage 3 AKD) is defined as an increase in serum creatinine level to ≥2.0 times that occurred or persisted within 90 days after renal injury. Abbreviation: AKD, acute kidney disease.

**Table 3 pharmaceuticals-16-01087-t003:** Factors associated with mean change in eGFR over time.

	Model 1			Model 2		
Variables	B	95% CI	*p* Value	B	95% CI	*p* Value
Treatment						
Sirolimus	−7.03	(−10.39, −3.68)	<0.0001	−4.37	(−6.89, −1.85)	0.0007
Mycophenolate mofetil	Ref			Ref		
TIME	−0.37	(−0.48, −0.25)	<0.0001	−0.46	(−0.54, −0.37)	<0.0001
Sirolimus * TIME	−0.12	(−0.28, 0.03)	0.1179	−0.18	(−0.30, −0.06)	0.0040
Mycophenolate mofetil * TIME	Ref			Ref		
Baseline proteinuria				−4.99	(−8.05, −1.92)	0.0016
HBV				−0.20	(−2.08, 1.68)	0.8306
HCV				1.97	(−0.12, 4.06)	0.0648
HCC				1.03	(−0.88, 2.94)	0.2884
AGE				−0.38	(−0.46, −0.29)	<0.0001
MELD				−0.12	(−0.24, 0.00)	0.0453
baseline eGFR group				18.67	(16.73, 20.61)	<0.0001
Tacrolimus 6 m trough				0.22	(−0.13, 0.57)	0.2077
Tacrolimus index date trough				0.90	(0.44, 1.36)	0.0001

Note: Time (follow-up time) is the number of 6-month intervals during the follow-up period. Abbreviations: HBV, hepatitis B virus; HCV, hepatitis C virus; HCC: Hepatocellular carcinoma; MELD: Model for End-Stage Liver Disease.

## Data Availability

Data supporting the reported results are available upon request from the corresponding author.

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
