# Peer review of "Impact of Sirolimus versus Mycophenolate Mofetil on Kidney Function after Calcineurin Inhibitor Dose Reduction in Liver Transplant Recipients"

_pharmaceuticals, 2023, doi:10.3390/ph16081087_

Round 1
Reviewer 1 Report
This work discusses the association between impaired kidney function and liver transplantation, highlighting the negative impact of immunosuppressant medications on kidney function. The study aimed to compare the kidney outcomes of patients treated with a reduced dose of tacrolimus combined with either sirolimus or mycophenolate mofetil (MMF). The results indicate that there were no significant differences in acute kidney disease risk between the two immunosuppressants. However, the use of sirolimus was found to be associated with a higher risk of a decline in estimated glomerular filtration rate greater than 30% compared to MMF. The study suggests that sirolimus use may lead to worse long-term kidney outcomes than MMF and emphasizes the importance of closely monitoring kidney function, adjusting medication dosage, and timely transitioning to MMF for liver transplant patients receiving sirolimus.
Overall, the study provides valuable insights into the effects of sirolimus and MMF on kidney outcomes in liver transplant patients.
it would be helpful to include a brief explanation of the standard dose of CNIs (calcineurin inhibitors) and their association with renal dysfunction for better context.
Considering previous studies on CNI dose reduction and its impact on renal function, the objectives of this study is underscored.
Address any potential confounding factors and the selected method for statistical methods used to compare the groups.
Please elaborate on the implications of this study findings in the context of previous studies and their impact on clinical decision-making.
The authors mention that the requirement for informed consent was waived due to the retrospective study design. It would be helpful to provide a brief justification for this decision
Provide detailed numbers excluded and the reason of exclusion. Mention the total number of patients initially screened and the final number included in the analysis.
The authors state that the tacrolimus dose was adjusted to maintain a trough level of 5-10 µg/mL during the first week. However, it would be helpful to specify the target trough level beyond the first week and any considerations for adjusting the dose based on patient factors or adverse effects.
The authors mention that basiliximab was routinely administered to the patients in the study as an induction therapy. It would be beneficial to provide a brief rationale for the use of basiliximab and any relevant evidence supporting its effectiveness in this context.
Discuss the rationale for selecting 30% and 50% decline in eGFR from baseline as outcomes of interest. Providing an explanation for these specific thresholds
In conclusions
Provide some recommendations or guidelines for clinicians regarding the frequency and methods of monitoring, as well as specific thresholds for dose adjustment or transition to MMF.
Consider adding a sentence or two discussing the limitations of the study, such as potential confounding factors or limitations in generalizability, to provide a balanced perspective on the findings.
Minor spell checks
Author Response
Overall, the study provides valuable insights into the effects of sirolimus and MMF on kidney outcomes in liver transplant patients. it would be helpful to include a brief explanation of the standard dose of CNIs (calcineurin inhibitors) and their association with renal dysfunction for better context.
Response:
Thank you for your insightful feedback on our study. We highly value your engagement with these critical issues. To address your points, we have amended our manuscript on page 2, lines 18-21 as follows: "Possible mechanisms of CNI nephrotoxicity involve afferent and efferent glomerular arteriole vasoconstriction, increased platelet aggregation, and the subsequent development of thrombotic microangiopathy within the glomerular arterioles [18]."
Considering previous studies on CNI dose reduction and its impact on renal function, the objectives of this study is underscored. Address any potential confounding factors and the selected method for statistical methods used to compare the groups.
Response:
Thank you for your comment. As we described in 4.3 Covariates and Supplemental Table S1, known potential confounders associated selection bias and kidney outcomes were included in the study (page 11, line 10-19). We used propensity score matching to minimize selection bias as described in 4.4 Statistical Analysis section. In light of your suggestion, we are incorporating additional confounding factors such as blood loss, cold and warm ischemic time, and post-LT acute rejection rate in section 2.2 Intra- and post-operative factors (page 4, line 7-23).
We have also addressed more confounding factors in our revised manuscript in Discussion (page 9, lines 40-42) as follows: " Third, we acknowledge that unmeasured confounders such as volume status, smoking, socioeconomic status, and donor factors, associated with AKD and CKD development, were not assessed."
Please elaborate on the implications of this study findings in the context of previous studies and their impact on clinical decision-making.
Response:
We have added sentences in Discussion to elaborate both clinical decision-making and research implications of the study results as follows:
“These study results highlight the need of tailored immunosuppression for LT recipients who are at risk of kidney disease. Use of sirolimus is not preferable for patients with existing kidney disease such as previous AKD and MMF combination may be a better strategy for long-term kidney protection. Monitor kidney function and trough level of tacrolimus and sirolimus are imperative for the protection of kidney function. Furthermore, these study findings can serve as a platform for future research comparing the effectiveness and safety between different strategies of immunosuppression.” (page 9, line 48-55)
The authors mention that the requirement for informed consent was waived due to the retrospective study design. It would be helpful to provide a brief justification for this decision
Response:
Thanks for mentioning this important issue. To avoid repeating statement, we had omitted the sentence “The requirement for informed consent was waived because of the retrospective study design” in the Methods section and added a brief statement on page 12, line 8-9 as below:
“The requirement for informed consent to individual patients was waived because this is a retrospective observation study with de-identified patient-level record.”
All study analyses were conducted based on retrospective data with de-identified patient-level records, so informed consents to individual patients were waived by the Institutional Review Board of Chang Gung Memorial Foundation at Taipei, Taiwan.
Provide detailed numbers excluded and the reason of exclusion. Mention the total number of patients initially screened and the final number included in the analysis.
Response:
Thank you for your comment. We have provided details of patient inclusion and exclusion criteria. The revised content in our manuscript on page 2 line 53 - page 3 line 7 as follows:
"Between April 2011 and December 2017, 789 patients underwent Liver Transplantation (LT) at Kaohsiung Chang Gung Memorial Hospital (KCGMH). We identified 472 patients who met the inclusion criteria (318 in the sirolimus group and 154 in the MMF group). Notably, 69 patients were excluded due to a follow-up period of less than 3 years, of which 41 were foreign nationals and could not be consistently monitored at our center. Furthermore, 74 patients were excluded because they did not use sirolimus or MMF as their immunosuppressant. An additional 52 patients were excluded as they were treated with counterpart therapy during the follow-up period. Finally, 5 patients were excluded as they used the target agents for less than a year (Figure 1)."
The authors state that the tacrolimus dose was adjusted to maintain a trough level of 5-10 µg/mL during the first week. However, it would be helpful to specify the target trough level beyond the first week and any considerations for adjusting the dose based on patient factors or adverse effects.
Response:
Thank you for your inquiry. We apologize that we did not make it clear and have clarified the treatment goal of tacrolimus in accordance with our institution's Living Donor Liver Transplant (LDLT) protocol. The details have been amended in our manuscript on page 10, lines 11-16 as follows: "Administration of tacrolimus is postponed until improvement in renal function. Upon achieving these markers, oral tacrolimus administration is initiated at 0.15 mg/kg/day. Dosage adjustments are made to achieve a trough level of 10–15 ng/ml during the first week, >6 ng/ml beyond the first week, and >4 ng/ml after the first year, or even <4 ng/ml, provided there is normal liver function [47]." The dose and trough level of sirolimus have been amended on page 10, line 20-21 as below: “Sirolimus was initiated at 1mg once daily for 3-7 days and then titrated to main a trough level of 4-10 ng/m [49].”
The authors mention that basiliximab was routinely administered to the patients in the study as an induction therapy. It would be beneficial to provide a brief rationale for the use of basiliximab and any relevant evidence supporting its effectiveness in this context.
Response:
Thank you for your comment. We have provided more detailed information regarding the clinical use of basiliximab in our center in the revised manuscript on page 10, lines 5-11:
" We routinely injected basiliximab (20 mg), a CD25 monoclinal antibody, 6 h after portal vein reperfusion and on post-transplant day 4 for induction therapy. The implementation of basiliximab has been improved with reduced biopsy-proven acute rejection rates during the initial 6 months and improved problem-free post-LT survival over the first 12 months. Additional advantages of basiliximab include the preservation of kidney function, reduction in steroid-resistant acute rejection occurrences, and lower incidence of post-transplant diabetes mellitus [46]."
Discuss the rationale for selecting 30% and 50% decline in eGFR from baseline as outcomes of interest. Providing an explanation for these specific thresholds
Response:
Thank you for your comment. We have revised our manuscript to emphasize the importance of a 30% and 50% decline in eGFR on page 6, lines 7-12 as follows: "Decreases in estimated GFR smaller than a doubling of serum creatinine concentration is strongly linked to the risk of End-Stage Renal Disease (ESRD) and mortality, supporting the consideration of lesser declines in estimated GFR (such as a 30% reduction over 2 years) as an alternative endpoint for CKD progression [28]. Moreover, a 50% decline in eGFR is often regarded as a major indicator of adverse kidney events in many significant studies [29, 30]"
In conclusions
Provide some recommendations or guidelines for clinicians regarding the frequency and methods of monitoring, as well as specific thresholds for dose adjustment or transition to MMF.
Response:
As suggested, we have added policy implications in conclusions to highlight the importance of the study results as follows: " These findings highlight the needs of periodical monitoring of kidney function and early detection of kidney injury at least every 1-3 months; other reno-protective strategies, such as dose optimization of immunosuppression and early transition to MMF in LT patients using sirolimus.” (page 11, line 46-49)
Consider adding a sentence or two discussing the limitations of the study, such as potential confounding factors or limitations in generalizability, to provide a balanced perspective on the findings.
Response:
Thank you for your comments. We have added more information related to operation such as blood loss, cold and warm ischemic time in section 2.2 Intra- and post-operative factors (page 4, line 7-23). We have also revised our manuscript on page 9, lines 40-42 as follows: " Third, we acknowledge that unmeasured confounders such as volume status, smoking, socioeconomic status, and donor factors, associated with AKD and CKD development, were not assessed. "

Reviewer 2 Report
This study aimed to compare the effect of the sirolimus versus mycophenolate mofetil in liver transplant patients, both group with a reduced dose od tacrolimus, looking at outcomes on renal function.
Major revision:
- What’s the rejection rate in both group?
- The authors should take in account also other risk factors for AKI in liver transplant patients:
ïƒ pre-transplant: incidence of NASH, autoimmune diseases (PSC, PBC), metabolic syndrome before the transplant.
ïƒ Intraoperative risk factors: blood loss, CIT, WIT
ïƒ immediate post-transplant liver function
-
Minor revisions:
- Were steroids used routinely?
Author Response
Reviewer 2
What’s the rejection rate in both group?
Response:
Thank you for your constructive suggestion. By reviewing the patients' charts, the acute rejection rate was not significantly different between the two groups and we have added this information into our manuscript on page 4, line 21-23: " The number of biopsy-proven acute rejection occurring in two months after LT were 15 in the sirolimus group (12.9%) and 11 (9.4%) in the MMF group (p=0.40)."
The authors should take in account also other risk factors for AKI in liver transplant patients:
- pre-transplant: incidence of NASH, autoimmune diseases (PSC, PBC), metabolic syndrome before the transplant.
- Intraoperative risk factors: blood loss, CIT, WIT
- immediate post-transplant liver function
Response:
Thank you for your comments. We agree these factors may also cause AKI. For pre-operational indication for liver transplantation, we have added sentences on page 3, line 21-24 as below.
“Few primary biliary cholangitis (PBC) was noted the two groups (3 patients in the sirolimus group and 5 in the MMF group (P=0.36)). Neither nonalcoholic steatohepatitis (NASH) nor primary sclerosing cholangitis (PSC) were observed in our study cohort.” For metabolic syndrome, we have already incorporated factors such as HTN, DM, and BMI into our PS matching.
For the operation-related factors that may affect renal outcomes, we have added on new section 2.2 for more details in our manuscript on page 4, lines 8-14 as follows: "Certain intra- and post-operative factors, such as blood loss, cold and warm- ischemic time and post-op acute rejection rate, may influence the renal outcomes after liver transplantation. There was no significant statistical difference between the two groups in terms of blood loss (sirolimus vs. MMF group: 3363 ± 3702 vs. 3178 ± 3720 ml, p = 0.71), warm ischemia time (sirolimus vs. MMF group: 42.21 ± 29.01 vs. 39.57 ± 18.85 min, p = 0.41), and cold ischemia time (sirolimus vs. MMF group: 42.47 ± 14.13 vs. 41.72 ± 11.09 min, p = 0.65) during surgery."
For postoperative liver function, because of large variation immediate after liver transplantation, we compared the liver function one month post-operation between the two groups. We have added more information on page 4, line 15-21:
“Liver function one month after LT were also compared between the two groups. There was no significant statistical difference in aspartate aminotransferase (AST) (sirolimus vs. MMF group: 42.49 ± 35.67 vs. 40.16 ± 27.71 U/L, p = 0.58), alanine transaminase (ALT) (sirolimus vs. MMF group: 66.87 ± 74.83 vs. 67.46 ± 62.98 U/L, p = 0.95), total bilirubin (sirolimus vs. MMF group: 0.96 ± 1.66 vs. 0.83 ± 0.42 mg/dl, p = 0.42), and albumin (sirolimus vs. MMF group: 3.74 ± 0.52 vs. 3.75 ± 0.44 g/dl, p = 0.41) between the two groups."
Minor revisions:
Were steroids used routinely?
Response:
Thank you for your comment. Steroids indeed are included in our immunosuppressive regimen. We have clarified this in our manuscript on page 10, lines 22-27 as follows: “Methylprednisolone (20 mg/kg IV) was administered, followed by a post-transplantation IV dosage of 2 mg/kg/day. This was gradually reduced to an oral prednisone dosage of 20 mg daily by the seventh postoperative day, eventually achieving a minimum dosage of 5 mg per day. From the third month post-transplantation, we began weaning patients off steroids, unless they have experienced rejection episodes or if the transplantation was performed due to an autoimmune disease.”

Reviewer 3 Report
Reviewer comments and suggestions
The authors in this study compared adverse kidney outcomes among patients treated with a reduced dose of tacrolimus (calcineurin inhibitor) plus sirolimus or mycophenolate mofetil (MMF) in the liver transplant center at Kaohsiung Chang Gung Memorial Hospital.
The study results reported that acute kidney disease risks in this study cohort were not significantly different for the two immunosuppressants. However, sirolimus use was significantly associated with a higher risk of estimated glomerular filtration rate decline > 30% than MMF. Therefore, the authors suggested that sirolimus use might have worsened long-term kidney outcomes compared to MMF.
Overall, the manuscript was well written. However, a few concerns/comments needed to be explained/modified.
- Its important data, please elaborate few more studies on the same “In a previous study, up to 20% of the recipients developed stage 4 or 5 CKD within 20 years of LT [4].”
- I saw their explanation of defining in other sections for AKI and CKD, it is better to explain in the introduction section
- What does it mean, please explain the line “many centers still use sirolimus to spare CNI use but avoid de novo use immediately after LT.”
- Please add up few references for these lines “Many studies have shown that MMF is a safe immunosuppressant that offers metabolic and renal”
- Comments for figures 2 and 3 Figures are not clear, please add with another one.
- I think the points should be earlier written “There were also no significant differences between the two groups in terms of age, sex, baseline renal function, or presence of comorbidities, including HBV, HCV, HCC, or DM”
- So the authors suggested that MMF was better “t MMF had a lower incidence of eGFR decline by > 30% than sirolimus (aHR, 2.09; 95% CI: 1.33–3.28, p < 0.0014) after 3 years of follow-up”
- It would be nice if the authors could add up tables or figures while discussing, easy to understand their points
- No need to cite two times the same references “No need to cite two times the same references”
- Comments for conclusion: Still the authors need to clear their conclusion based on the observed study
Author Response
Reviewer 3
The authors in this study compared adverse kidney outcomes among patients treated with a reduced dose of tacrolimus (calcineurin inhibitor) plus sirolimus or mycophenolate mofetil (MMF) in the liver transplant center at Kaohsiung Chang Gung Memorial Hospital. The study results reported that acute kidney disease risks in this study cohort were not significantly different for the two immunosuppressants. However, sirolimus use was significantly associated with a higher risk of estimated glomerular filtration rate decline > 30% than MMF. Therefore, the authors suggested that sirolimus use might have worsened long-term kidney outcomes compared to MMF. Overall, the manuscript was well written. However, a few concerns/comments needed to be explained/modified.
1. Its important data, please elaborate few more studies on the same “In a previous study, up to 20% of the recipients developed stage 4 or 5 CKD within 20 years of LT [4].”
Response:
Thank you for your compliments and comments. We have incorporated more references into our manuscript on page 2, lines 1-2. “In previous studies, up to 20% of the recipients developed stage 4 or 5 CKD within 20 years of LT [4,5].”
2. I saw their explanation of defining in other sections for AKI and CKD, it is better to explain in the introduction section.
Response:
Thank you for your suggestion. We have added the definition of AKI and CKD in the introduction section on page 2, line 5-8.
“AKI is a condition defined by a sudden loss of kidney function on the basis of increased serum creatinine levels or reduced urinary output within a duration of 7 days [6]. CKD is defined as abnormality of kidney structure or function for more than 3 months [7].”
3. What does it mean, please explain the line “many centers still use sirolimus to spare CNI use but avoid de novo use immediately after LT.”
Response:
Thank you for your comments. It has been reported that de novo use of sirolimus after LT is associated higher rate of hepatic artery thrombosis and graft loss, and hence FDA had once issued a black box (now has been removed). These effects can be mitigated by later use of sirolimus. Besides, sirolimus is more affordable than another mTOR inhibitor, everolimus. Given this, many centers continue to employ sirolimus as a means of sparing the use of CNI, but typically avoid its de novo use immediately following LT. Therefore, it is mentioned in our manuscript on page 2, line 28-31: “Hence, although the FDA issued a black box warning for increased hepatic artery thrombosis and graft loss in de novo liver transplant recipients [22], many centers still use sirolimus to spare CNI use but avoid de novo use immediately after LT.”
4. Please add up few references for these lines “Many studies have shown that MMF is a safe immunosuppressant that offers metabolic and renal”
Response:
Thank you for your comments. We have added more references to support this statement on page 2, line 38.
5. Comments for figures 2 and 3 Figures are not clear, please add with another one.
Response:
Thanks for your comments. We have enlarged the Figure 2 and 3 to make it clearly seen.
6. I think the points should be earlier written “There were also no significant differences between the two groups in terms of age, sex, baseline renal function, or presence of comorbidities, including HBV, HCV, HCC, or DM”
Response:
Thanks for your comments. We suppose you are referring to figure 3a. We have revised our manuscript on page 5, line 35-37: “In subgroup analysis, there were also no significant differences between the two groups in terms of age, sex, baseline renal function, or presence of comorbidities, including HBV, HCV, HCC, or DM (Figure 3a).”
7. So the authors suggested that MMF was better “t MMF had a lower incidence of eGFR decline by > 30% than sirolimus (aHR, 2.09; 95% CI: 1.33–3.28, p < 0.0014) after 3 years of follow-up”
Response:
Well, from the results of the current study, we would say it looks like this. However, as we already mentioned in our manuscript, there are certain strengths (relative large sample size, PS matching to balance the difference between two groups..) and limitations in this study. Hence, we have added some suggestions to the readers:
“These study results highlight the need of tailored immunosuppression for LT recipients who are at risk of kidney disease. Use of sirolimus is not preferable for patients with existing kidney disease such as previous AKD and MMF combination may be a better strategy for long-term kidney protection. Monitor kidney function and trough level of tacrolimus and sirolimus are imperative for the protection of kidney function. Furthermore, these study findings can serve as a platform for future research comparing the effectiveness and safety between different strategies of immunosuppression.” (page 9, line 48-55)
8. It would be nice if the authors could add up tables or figures while discussing, easy to understand their points.
Response: Thank you for your suggestions. We have marked the table or figures in the discussion section.
9. No need to cite two times the same references “No need to cite two times the same references”
Response: Thank you for your suggestion. We have removed the reference cited for more than 2 times.
10. Comments for conclusion: Still the authors need to clear their conclusion based on the observed study
Response: Thank you for your suggestions. We have revised our conclusion into a more conservative saying and also added some suggestions on page 11, line 44-49:
“In conclusions, our study showed that, among LT patients, the use of sirolimus plus tacrolimus may carry a higher risk of long-term kidney function deterioration than the use of MMF plus tacrolimus. These findings highlight the needs of periodical monitoring of kidney function and early detection of kidney injury at least every 1-3 months; other reno-protective strategies, such as dose optimization of immunosuppression and early transition to MMF in LT patients using sirolimus.”

Round 2
Reviewer 1 Report
Accept in the current form